# Labeling T Cells to Track Immune Response to Immunotherapy in Glioblastoma

**John Y. Rhee** [1,2,*], **Jack Y. Ghannam** [3], **Bryan D. Choi** [3] and **Elizabeth R. Gerstner** [1]

1 Department of Neuro-Oncology, Massachusetts General Hospital Cancer Center, Boston, MA 02114, USA
2 Department of Neuro-Oncology, Dana Farber Cancer Institute, Brigham and Women's Cancer Center, Boston, MA 02215, USA
3 Department of Neurosurgery, Massachusetts General Hospital Cancer Center, Boston, MA 02114, USA
* Correspondence: jrhee@mgh.harvard.edu

**Abstract:** While the advent of immunotherapy has revolutionized cancer treatment, its use in the treatment of glioblastoma (GBM) has been less successful. Most studies using immunotherapy in GBM have been negative and the reasons for this are still being studied. In clinical practice, interpreting response to immunotherapy has been challenging, particularly when trying to differentiate between treatment-related changes (i.e., pseudoprogression) or true tumor progression. T cell tagging is one promising technique to noninvasively monitor treatment efficacy by assessing the migration, expansion, and engagement of T cells and their ability to target tumor cells at the tumor site.

**Keywords:** T cell tagging; immunotherapy; glioblastoma





## 1. Introduction

Immunotherapy has revolutionized cancer treatment by leveraging the immune system's capacity to kill neoplastic cells for therapeutic purposes. One major class of therapy to be approved for cancer was immune checkpoint inhibitors (ICI). These agents target co-inhibitory axes involving PD-1, PDL-1, and CTLA-4 that normally dampen the immune system's response to antigens. This can lead to a T cell-mediated immune response against tumors, but at the same time, mediate immune-related adverse events (irAEs) affecting normal tissue. In addition to ICIs, there have been a host of other immunotherapies, including cancer vaccines, chimeric antigen receptor (CAR) T cell therapy, tumor microenvironment modulators, and other immunomodulatory agents to rev up the immune system's response to cancer [1]. Though immunotherapy, especially ICIs, has borne considerable success in some solid tumors, such as advanced melanoma and non-small-cell lung cancer, these successes have thus far not been replicated in the treatment of glioma; the reasons for this inherent resistance are yet to be fully elucidated.

## 2. Immunotherapy for Gliomas

Over the last decade, two seminal studies discovered a network of functional lymphatic vessels that line the dural sinuses, draining into the deep cervical lymph nodes. It is now thought that these lymphatic vessels serve as a conduit for the trafficking of immune cells, such as T cells, between the peripheral immune system and the central nervous system (CNS)—a mechanistic suggestion that immunotherapy may prove efficacious in treating CNS malignancies [2]. Consequently, several studies have explored various immunotherapies in glioblastoma (GBM), including ICI, CAR T cells, and cancer vaccines.

Though pre-clinical work for ICI use in GBM was promising, clinical trials have proven disappointing with no overall survival benefit [3]. For example, there was no survival benefit of nivolumab (an anti-PD-1 agent) over bevacizumab or in adding nivolumab to radiation with or without temozolomide in newly diagnosed GBM [4,5]. This failure of ICIs to demonstrate significant benefit in GBM may be due to various reasons that include the

blood brain barrier (BBB), which limits delivery of larger antibody-based or water soluble drugs; an immunologically "cold" tumor microenvironment with little T cell infiltration, but instead infiltration by myeloid-derived suppressors cells and regulatory T cells that suppress effector T cell activity; and an abnormal tumor vasculature that promotes an hypoxic tumor microenvironment (TME).

CAR T cells have shown striking clinical response in some hematologic malignancies, and therefore, there has been significant interest in extending their use for solid tumors. While still an active area of clinical study in GBM, target tumor antigens have included EGFR/EGFRvIII, IL13Ra2, and HER2 [6–8]. To date, however, CAR T cells have demonstrated limited efficacy for brain tumors in general, with various aspects of underlying brain tumor immunobiology confounding effect treatment, including the lack of highly and uniformly expressed tumor antigens, limited CAR T cell trafficking to the tumor, tumor antigen loss, and an immunosuppressive tumor microenvironment.

Cancer vaccine trials have also shown proof-of-concept feasibility and efficacy, utilizing vaccines designed to target neoantigens found in patients with glioma-bearing EGFRvIII [9], isocitrate dehydrogenase-1 (IDH1) mutations [10], as well as other tumor antigens associated with GBM. Patients are vaccinated with dendritic cells loaded with tumor-associated antigens, which migrate to local lymph nodes, present these antigen-derived peptides on human leukocyte antigen (HLA) molecules, and initiate an antitumoral T cell response, selectively killing tumor cells and preventing tumor recurrence due to immunological memory. However, multiple randomized trials have shown mixed results, and further studies are underway to try to improve target antigen selection, cell preparation, and integration of cancer vaccines with other treatment regimens [11–13].

One clinical conundrum with immunotherapy is that there may be an initial increase in tumor size (pseudoprogression) before regression occurs, with tumor shrinkage as a later endpoint. Within the fixed volume associated with having a skull, this increase in size can lead to neurological decline and need for steroids to control edema, thus, dampening the immune response. Distinguishing between pseudoprogression and true progression is important for clinical decision-making and whether to continue a certain treatment. Unfortunately, there is no perfect diagnostic tool that can reliably distinguish between the two entities as even biopsy is subject to sampling error.

### 3. Importance of T Cells

One of the main goals for immunotherapeutic approaches is to enhance cytotoxic T lymphocyte infiltration and effector function to augment endogenous antitumor control. In the cancer-immunity cycle, endogenous antitumor immune responses initiate at the tumor where activated tissue-resident antigen-presenting cells (APCs) phagocytose tumor–cell debris and migrate to secondary lymphoid organs (SLOs). In the SLOs, activated APCs present tumor antigen on major histocompatibility complex (MHC) class I and II, meeting naive T cells which continually circulate between SLOs "in search" for their cognate antigens. When T cell-receptor (TCR)-specific antigens are recognized, T cell populations expand and egress from the SLOs and infiltrate the TME with their resultant cytotoxic functions as effector T cells. When successful, this immune response results in tumor destruction with release of more tumor antigens, inducing greater influx of effector cells, and continuing the cancer–immunity cycle. However, when chronically exposed to their cognate antigen, such as in cancer or chronic infection, T cells adopt an "exhausted" phenotype, marked by upregulated expression of checkpoint molecules (e.g., CTLA-4 and PD-1), which evolved to promote the preservation of a balance between cytotoxicity and host tissue integrity.

### 4. T Cell Labeling

Given the importance of T cells as an effector arm of the immune system, there is a great deal of interest in better understanding their role in mediating a successful response to immunotherapy. The ability to determine in vivo the location, distribution,

and long-term viability of cell populations, and their biologic fate with respect to cell activation and differentiation, is referred to as cell-tracking [14]. Cell tracking involves non-invasive methods for monitoring the distribution and migration of biologically active cells in living organisms by pairing imaging modalities and cell-labeling methods, which allows for visualization of labeled cells in real time, as well as monitoring and quantifying cell accumulation and function.

While labeling T cells is relatively straightforward in the laboratory, it becomes more complex in humans. In humans, cell-tracking can be divided into ex vivo and in vivo labeling. The difference between ex vivo and in vivo labeling is that ex vivo labeling happens outside of the body whereas in vivo labeling involves in situ imaging of cells by injecting radioactive, fluorescent, or luminescent tracers or antibodies.

Ex vivo cell labeling classically involves removing T cells and then labeling them intracellularly or on the cell membrane with long-lived radionuclides or other contrast agents such as iron oxides before re-injection [14,15]. These techniques have been used for many years, but there are concerns about cell viability and maintenance of cell function. In vivo tracking can be done by either direct cell labeling, where a contrast agent is directly loaded into the therapeutic cells, or indirect cell labeling, which relies on genetic engineering of the cell-based therapeutic to express a reporter gene that enables contrast formation upon administration of a contrast agent (Figure 1). This labeling process can consist of genetically engineering the expression of proteins, allowing uptake of an imaging agent. In addition, techniques such as metabolic engineering or click chemistry function [16] at the cell membrane, taking advantage of fast and high-yield chemical reactions that take place in aqueous media or in vivo [14]. Other techniques include targeting through labeled peptides or antibodies [17] that bind to the cell of interest or a small, labeled probe targeting an antibody or binding protein [14].

Integral to those methods of T cell tracking that require leveraging known T cell surface molecules is flow cytometry. Specifically, T cells represent a functionally and developmentally heterogenous population of immune cells that have historically been identified and characterized by positively and negatively selecting for various clusters of differentiation (CDs). These CDs encompass cell-surface molecules that represent functional markers of specific T cell subsets (e.g., CD8 as a co-receptor of TCR engagement), markers of differentiation state, chemokine receptors, and various extracellular cytokines, among others. Intracellular components such as transcription factors and cytokines can additionally be stained to profile T cell subsets. Given the high dimensionality of immunophenotyping that flow cytometry affords, this technique has enabled precise identification and quantification of T cell subsets and their respective markers. These features of flow cytometry are particularly salient in the context of tracking T cells for tumors such as GBM, where different T cell subtypes are known to play different pathologic and prognostic roles. For example, tumor-infiltrating FOXP3$^+$ regulatory T cells (Tregs), whose canonical function is immunosuppressive, have been shown to be associated with reduced survival and tumor recurrence in patients with GBM, while the opposite is true for cytotoxic CD8$^+$ T cells. Thus, parallel tracking of subsets with opposing roles in pathology may offer non-invasive, multi-dimensional, and orthogonally validated prognostication for patients. Flow cytometry also enables identification of rarer subsets of T cells for tracking, including neo-antigen specific T cells and various memory T cell subtypes (e.g., central versus effector memory); investigation of such subsets for T cell trafficking may enable a more nuanced characterization of how a patient's disease is responding to treatment, and evolving over time. However, it must be recognized that there are several technical limitations of flow cytometry when it is utilized for identifying markers for T cell tracking in research and in the clinical setting. These include limits on the number of markers that can be stained at once, the fact that analyzed cells must be in suspension, high-level training requirements to perform the test accurately, and that cells must be viable to analyze properly. Nevertheless, given the spectrum of roles that the T cell subsets play in cancer, flow cytometry plays an invaluable role in identifying sensitive, specific, and functionally inert markers for T cell tracking.

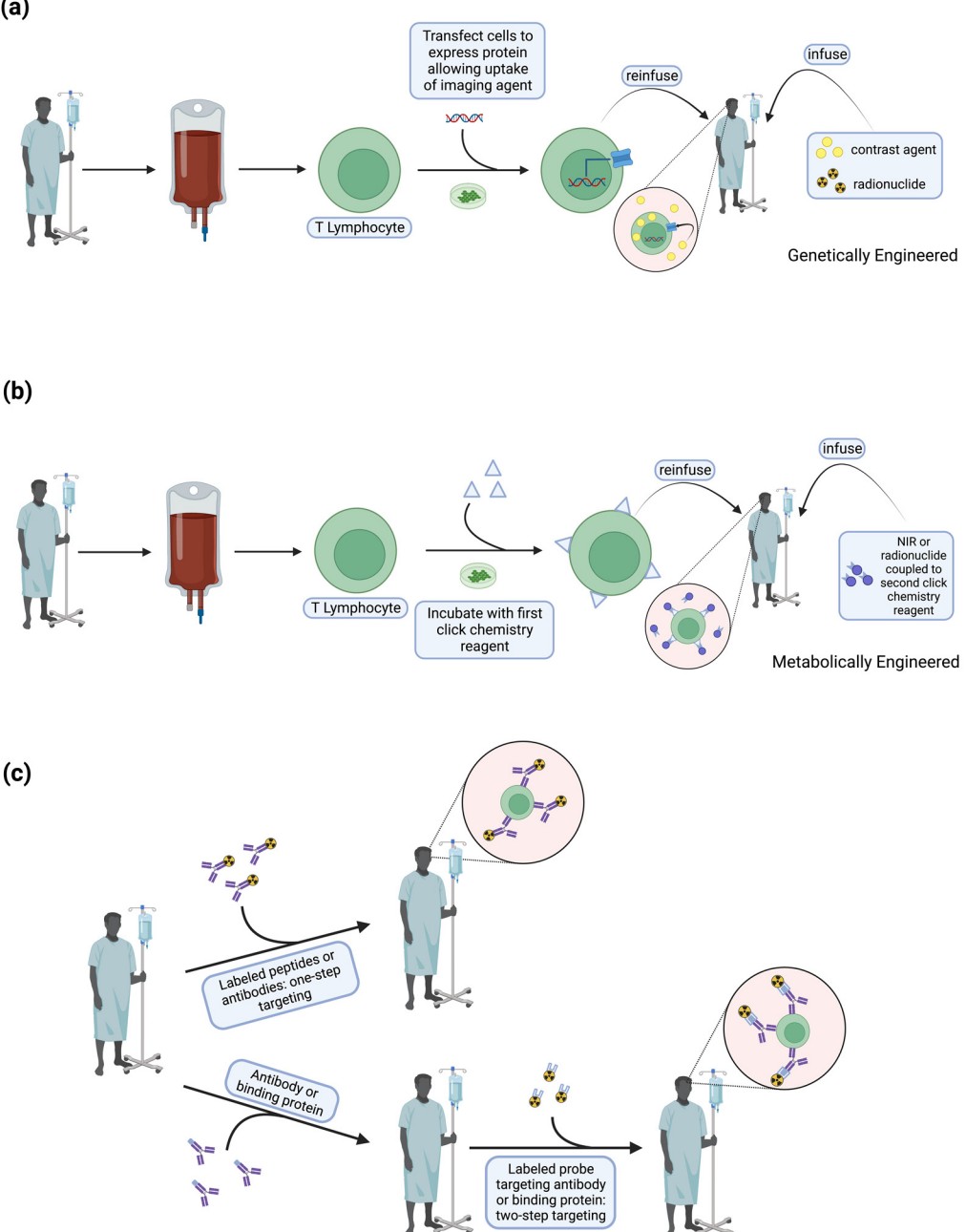

**Figure 1.** Schematic illustrating three broad approaches of in vivo T cell labeling. (**a**) Method of genetically engineering T cells to express proteins that allow uptake of separately infused imaging agents. (**b**) Method of metabolically engineering T cells to express a click chemistry small molecule on the cell surface, enabling bioconjugation of a separately infused radionuclide or near-infrared (NIR) fluorophore that is coupled to a second click chemistry small molecule. (**c**) Illustration of one- and two-step targeting. In one-step targeting, labeled peptides or antibodies conjugated to a radionuclide or MRI contrast agent are infused and bind to cognate antigen on the T cell surface in vivo. In two-step targeting, a binding protein or antibody is infused separately from a small, labeled, probe, which is able to target the binding protein or antibody in vivo.

T cell labeling will be critical to identify where T cells are homing, if T cells are active, and if there is replication or persistence at the tumor site. Quantification of T cell infiltration in the tumor and local anti-tumor effector functions would help clinicians define thresholds

for successful effect of immunotherapy at earlier stages during treatment and improve our understanding of why immunotherapy is (or is not) working in tumors.

## 5. Animal Studies

The most common approaches to labeling T cells or measuring T cell activation status has involved radioactive tracers and PET imaging, although SPECT has been used as well but is limited by its poorer sensitivity and spatial resolution [18] (summary of T cell markers in Table 1). A [64]Copper(Cu)-labeled diabody specific for CD8 has been used to assess CD8 T cell density and treatment-related changes [19] in tumors in mice, and whole antibodies or antibody fragments to CD3, CD4, and CD8 labeled with [89]Zirconium(Zr) have also been used [20,21]. Antibody fragments may be of higher clinical utility than full antibodies, as the former reach their target faster while full antibodies accumulate slowly in peripheral tissues, thus requiring imaging a day or days after tracer administration.

**Table 1.** Select animal studies targeting T cell markers.

| Label | Target | Imaging Modality | Cancer |
| --- | --- | --- | --- |
| [64]Copper-labeled diabody [19] | CD8 | PET | Her2 breast cancer |
| [89]zirconium-labeled antibody [20,21] | CD3 | PET | Colon cancer, bladder cancer |
| [18]fluorobenzoyl-interleukin-2 radiotracer | CD25 | PET | Cervical cancer |
| [64]Copper-conjugated murine Ab specific for OX40 receptor [19] | OX40 receptor | PET | lymphoma |
| [89]Zr-oxinate [22] | CART | SPECT | Breast cancer, myeloma, glioblastoma |
| Ferucarbotran [23,24] | CART | Magnetic particle imaging | Glioblastoma |
| Granzyme B (GZP—peptide in PET imaging) | Granzyme B | GZP PET signal | Colon carcinoma |

Other targets have included an [18]fluorobenzoyl-interleukin-2-labeled tracer and T cell activation through targeting the activation marker OX40 with a [64]Cu-conjugated murine antibody specific for the OX40 receptor, which is upregulated on the surface of T cells upon antigen-specific activation [25]. A PET imaging agent targeting a functional marker of effector T cell activation, granzyme B, allows for direct quantification of anti-tumor response before changes in tumor volume. Granzyme B is in cytotoxic granules of T cells and gest released as both intracellular and extracellular and, with a biological half-life of 14 days, it is a stable target for immune activation. Using a specific peptide PET imaging agent for granzyme B (GZP), authors of one study showed that high GZP PET signal predicted response to therapy and low signal predicted progression, with a sensitivity of 93% and negative predictive value of 94%.

T cell receptors (TCRs) are also an attractive group of imaging targets due to continuous T cell membrane turnover resulting in TCR internalization, and therefore accumulation of tracer within the cells. A [64]Cu-labeled anti-chicken OVA-TCR antibody showed efficient internalization within thirty minutes, and a [89]Zr-labeled anti-mouse-TCR F(ab')2 fragment has also been studied [26].

There have also been advances in direct in vivo imaging. Longer half-life isotopes such as [89]Zr-oxine in PET imaging show promise, and in xenograft mouse models of glioblastoma [89]Zr-oxine-labeled CAR T cells were detectable for up to six days [22]. Chelators directly bound to cell-surface proteins circumvent disruption of the plasma membrane. [89]Zr-desferrioxamine-NCS-conjugate amines on cell-surface proteins have been shown to be retained for up to seven days without affecting cell viability [27].

[89]Zr derivatives have also been utilized for in vivo tracking by leveraging [89]Zr-labeled F(ab')2 targeting various T cell markers; for example, CD7, a marker of mature T cells, has been shown in pre-clinical studies to provide a robust signal at the tumor site without impacting T cell function nor tumor rejection. Anti-CD2 [89]Zr antibody conjugates have also been explored for T cell tracking given the marker's expression highly correlates with T cell cytolytic activity within tumors; however, in vivo studies demonstrated targeting this marker induces significant T cell depletion with subsequent failure of tumor rejection,

highlighting the importance of identifying clinically relevant but functionally inert markers. Other antibody-conjugated [89]Zr tracers that have been studied include those targeting CD8 and CD3.

Another in vivo imaging method is magnetic particle imaging, a noninvasive technique separate from magnetic resonance imaging (MRI) that directly detects superparamagnetic nanoparticles and has been used to monitor transplantation, bio-distribution, and clearance of ferucarbotran-labeled human stem cells [23]. Ferucarbotran (Resovist), a superparamagnetic iron oxide approved by the FDA, has also been used to track long-term fate of in vivo neural cell implants [24].

MRI-based contrast agents have also been explored using gadolinium, ferumoxytol, or other nanoparticles, or superparamagnetic iron oxide (SPIO) to label different immune cells [28]. [19]Fluorine(Fl) MRI has been used to image activated T cells in vivo in mouse models over a 3-week period, though quantifying the amount of label at a site becomes less accurate over time due to cell division and loss of the label [29].

## 6. Human Studies

A few of these approaches have advanced to be used in humans (summary of T cell makers in Table 2). The cytokine IL-2 has been used as a marker for activated T cells by assessing expression of IL-2RA using SPECT imaging, specifically in metastatic melanoma (99 mTc). However, given radiolabeled IL-2 is a bioactive cytokine, one out of five patients in the study had infusion-related side effects [30]. Additional in vivo T cell surface markers are being explored to find the correct dosing for cepilimab in patients with advanced malignancies, such as using lymphocyte-activation gene 3 with [89]Zr-DFO for lymphoma [31].

**Table 2.** Select human studies targeting T cell markers.

| Label | Target | Imaging Modality | Cancer |
|---|---|---|---|
| [99]mTc [30] | IL-2 | SPECT/CT | Metastatic melanoma |
| [89]Zr-DFO [31] | Lymphocyte-activating gene 3 | PET | Lymphoma |
| [64]Copper-labeled antibody [32] | PD-1 | PET | Non-small-cell lung cancer |
| [89]Zirconium [32] | PD-1 IFN-gamma, IL13Rα2-targeted CAR T cells | PET | Non-small-cell lung cancer, breast cancer, glioblastoma |
| [18]Fluorobenzoyl-labeled clofarabine [33] | Enzyme in deoxycytidine kinase pathway | PET | Lymphoma |
| 2′-deoxy-2′-fluoro-9-B-arabinofuranosylguanine [33] | Enzyme in deoxyguanosine kinase pathway | PET | Acute graft versus host disease |
| 9-[4-[[18]F]fluoro-3-(hydroxymethyl)butyl]guanine ([[18]F]FHBG) [34] | CART engineered to express herpes simplex virus type-1 thymidine kinase (HSV1-TK) and interleukin-13 | PET | Glioma |

Tumor cells upregulate PD-L1 which binds to PD-1 on T cells and reduces T cell effector function and contributes to T cell "exhaustion." Monoclonal antibodies targeting the PD-1/PD-L1 axis have been explored, including [64]Cu-labeled anti-PD-1 antibodies [35] and an [89]Zr-labeled nivolumab in non-small-cell lung cancer patients [36]. Anti-IFN-γ [89]Zr-labeled probes for PET imaging have also been developed since IFN-γ plays an important role in the T cell signaling axis [37].

Activated T cells also switch on metabolic programs and upregulate the influx of substrates compared to non-active cells, and therefore, these metabolic pathways can be used to distinguish between active and non-active T cells. Tracers have been developed as substrates for key enzymes in the deoxyribonucleoside salvage pathway—specifically, deoxycytidine kinase (dCK) and deoxyguanosine kinase (dGK). Various tracers, including [18]Fluorobenzoyl-labeled clofarabine (a nucleotide purine analogue metabolized via dCK) [25] and 2′-deoxy-2′-fluoro-9-B-arabinofuranosylguanine (which accumulates in acti-

vated T cells via the dGK pathway) [33], have been developed to target these substrates, the latter having been tested in mouse models and humans.

Other in vivo imaging options are also available. T cells can be directly labeled by passive membrane diffusion, binding membrane molecules, or endocytosis, and then directly imaged based on the specific activity of the tracers and retention of radioactivity in cells. PET imaging using $^{64}$Cu-diethyldithiocarbamate and tropolonate has been used, though rapid efflux from cells and strong uptake in the liver restricts the use of $^{64}$Cu complexes in humans [32]. Limitations of direct labeling in patients include radio- or chemical toxicity related to the properties of the radionuclide, as well as potential longer-term toxicity related to leakage of long-lived radiotracers such as $^{89}$Zr.

Ex vivo approaches have focused on T cells which can be transduced to introduce genes for expression of CARs and TCRs (Figure 2). Reporter genes can be integrated within the T cell genome, so that imaging can continue in vivo as long as the cell therapy persists, and the reporter gene is passed on to, and maintained in, daughter cells upon cell division, allowing for expanding and contracting populations to be traced and measured over time. Limitations include cost and regulatory burden due to risks of aberrant viral integration [38]. However, this type of strategy can be employed to monitor the efficacy of therapies where cells are already being engineered such as in CAR T cell therapy. In fact, immunotherapy using CD8+ cytotoxic T lymphocytes (CTLs) engineered to express both herpes simplex virus type-1 thymidine kinase (HSV1-TK) and interleukin-13 zetakine CAR has been explored to treat high-grade gliomas, and PET imaging with a 9-[4-[$^{18}$F]fluoro-3-(hydroxymethyl)butyl]guanine ([$^{18}$F]FHBG) can be used to track *HSV1-tk* reporter gene expression in CAR-engineered T lymphocytes [34].

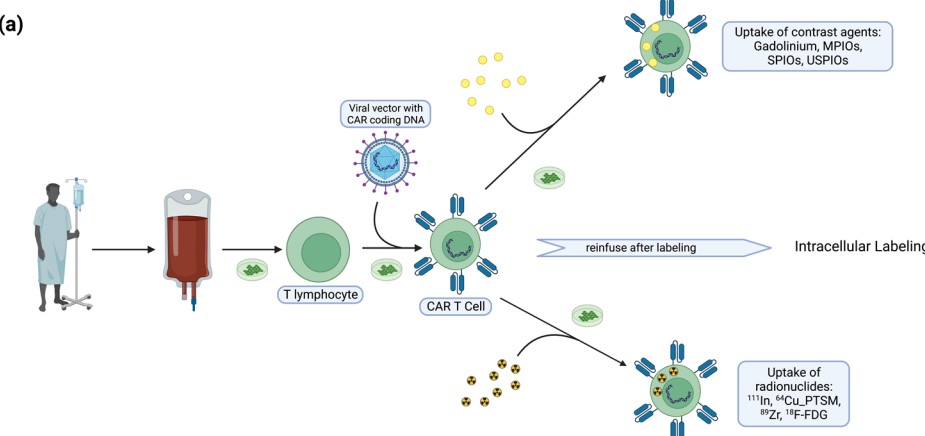

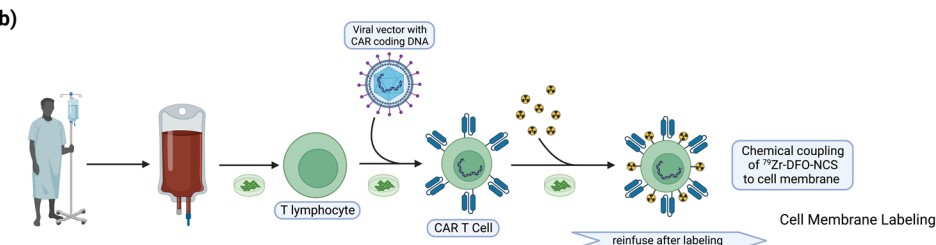

**Figure 2.** Schematic illustrating the ex vivo intracellular versus cell membrane labeling of CAR T cells. (**a**) Intracellular labeling of CAR T cells can be performed via ex vivo uptake of contrast agents or radionuclides by CAR T cells, which are subsequently infused in the patient. (**b**) Cell membrane labeling of CAR T cells can be performed by chemically coupling $^9$Zr-DFO-NCS to the cell membrane ex vivo, with subsequent infusion into the patient.

## 7. T Cell Tracking in Gliomas

Though there has been considerable effort in developing T cell tracking methods as described above, meaningful application of these techniques to gliomas has remained limited. Most of the work has been done in reporter genes, where CAR T cells are engineered such that their effectiveness and distribution can be tracked using HSV1-tk reporter gene expression [34]. One study used [$^{18}$F]FHBG to track engineered cytotoxic T lymphocytes to express IL-13 zetakine CAR and wild-type HSV1-tk gene reporter. This was conducted in a sample of seven patients with recurrent high-grade gliomas. The investigators were able to show that [$^{18}$F]FHBG accumulates in these modified T-lymphocytes expressing the reporter gene, and that accumulation of [$^{18}$F]FHBG at one hour was twelve-fold higher in transfected T lymphocytes compared to non-transfected cells. The investigators also showed that incubation for 60 min with [$^{18}$F]FHBG did not have any significant effect on cell proliferation, and including up to 48 h after the addition of [$^{18}$F]FHBG. Furthermore, there was no significant difference in normal brain uptake when comparing mean standardized uptake value (SUV) of [$^{18}$F]FHBG in non-transfected and transfected cells. [$^{18}$F]FHBG biodistribution showed maximum intensity 152 min after intravenous injection [34].

In mouse models of GBM, CAR T cells labeled with $^{89}$Zr-oxine showed that IL13R$\alpha$2-CAR T cells were labeled successfully without reduced efficacy from labeling and labeled CAR T cells were successful in assessing cytokine production and tumor cytotoxicity as well as in vivo antitumor activity [39]. CAR T cells labeled with $^{89}$Zr-oxine were injected intra-tumoral or intraventricularly. Activity of the CAR T cells was not affected when assessed with in vitro killing assays up to 145 h after labeling. Furthermore, labeling did not decrease IFN-gamma production of the labeled cells immediately after labeling and with sustained production 72 h after labeling. In addition, the $^{89}$Zr-oxine labeled CAR T cells had antitumor activity comparable to those of unlabeled CAR T cells [39].

Ferucarbotran-labeled pmel DsRed T cells against Kluc-gp100 GBM cells using magnetic particle imaging (MPI) have also been shown to be a potential viable marker in mouse models [40]. Investigators injected ferucarbotran-labeled DsRed T cells intracerebroventricularly in mice (n = 4), showing that these labeled T cells can be tracked in vivo using MPI. This signal decreased but persisted at 60 h post-injection. Ferucarbotran-labeled T cells were also seen in histologic sections of the brain after tail vein intravenous administration, showing entry into the brain. Lastly, the data showed that ferucarbotran labeling did not impair the ability of T cell activity, including measured production of IFN-gamma [40].

There are some challenges to T cell tracking in gliomas, including the fact that there is poor trafficking and persistence of T cells in gliomas and central nervous system malignancies. However, the development of these T cell tracking technologies would help to better understand the barriers to T cell persistence by allowing non-invasive and quantitative tracking of T cells over time [40].

Ideally, T cell tracers for diagnostic and trending purposes should not impact T cell function in any way, but there are risks of radionuclide-based impairments of T cell function. For example, an anti-murine CD4 cys-diabody has shown dose-dependent restrictions in T cell proliferation and IFN-gamma production (though not specifically in glioma). There are still very few studies that have been able to prove whether labeling T cells may diminish activity of T cells in human subjects, as compared to mice models. The opposite may also hold, where modulation of T cell function may lead to increased cytokine secretion after tracer binding and over-activation of T cells, resulting in cytokine release syndrome with theoretical risks of increased neurotoxicity. Further studies are needed to understand the impact of irradiation on various T cell subpopulations, and specifically how this may affect T cell tracking in gliomas.

## 8. Conclusions and Future Directions

Clinical application of T cell tracking for GBM is still in its infancy. As newer immunotherapy options continue to be tested in patients with GBM, T cell tracking will be important to directly monitor efficacy of immunotherapy to better understand why im-

munotherapy is (or is not) working and may help differentiate between pseudoprogression and true disease progression. While tumor biopsies with pathology are ideal for confirming response to immunotherapy, brain biopsies are invasive, subject to sampling error, or are not feasible in cases involving deeper structures. Therefore, advances in T cell tagging can help noninvasively monitor activity in immunotherapy.

As CAR T cell therapy is now being tested in early clinical trials for GBM, CAR T cells, which are already engineered ex vivo, can additionally be tagged with relatively little added manufacturing burden, holding great promise for monitoring treatment response by assessing the migration, expansion, and engagement of the CAR T cells and their ability to target cells at the tumor site. This way, by observing migration and engagement of CAR T cells, we can also predict those who may have a higher chance of responding to treatment. Misdiagnosis of pseudoprogression can lead to urgent treatment with high-dose steroids, which theoretically could decrease the effectiveness of CAR T cells. Tagging can help overcome this by providing an additional marker to assess for pseudoprogression and real progression of disease.

**Author Contributions:** Conceptualization, J.Y.R. and E.R.G.; writing—original draft preparation, J.Y.R.; writing—review and editing, J.Y.R., J.Y.G., B.D.C. and E.R.G.; visualization, J.Y.G.; supervision, E.R.G. All authors have read and agreed to the published version of the manuscript.

**Funding:** This research received no external funding.

**Institutional Review Board Statement:** Not applicable.

**Informed Consent Statement:** Not applicable.

**Data Availability Statement:** Not applicable.

**Conflicts of Interest:** The authors declare no conflict of interest related to this manuscript. The funders had no role in the design of the study; in the collection, analyses, or interpretation of data; in the writing of the manuscript; or in the decision to publish the results.

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
