# Peer review of "Labeling T Cells to Track Immune Response to Immunotherapy in Glioblastoma"

_tomography, doi:10.3390/tomography9010022_

Round 1

Reviewer 1 Report

The review article is about T-cell labeling to monitor and assess its therapeutic efficacy in cancer and its significance discusses in both mice and human studies. The authors have discussed T cell importance and labeling methods for t cells with an excellent graphical representation. However, in my opinion, the review article could have included additional details.

  1. Zr-89 is also used to trace other markers of T cells, for instance, CD2 and CD7 fab antibodies; these targets can be discussed in the review.
  2. In addition, labeling specificity, bioavailability, and tissue distribution of listed labeling agents in Tables 1 &2 can be included. The advantages and limitations of these agents could be discussed.
  3. T cells in the glioma section could be expanded. A table consisting of the T cell/T cell marker, labeling method, viability after labeling, functional properties of t cells, biodistribution, and anti-tumor properties for T cells in glioma sections can be included. Moreover, challenges in tracking T cells in glioma could be discussed.
  4.  A reference list could be included for the existing Table 1 and Table 2.

Author Response

The review article is about T-cell labeling to monitor and assess its therapeutic efficacy in cancer and its significance discusses in both mice and human studies. The authors have discussed T cell importance and labeling methods for t cells with an excellent graphical representation. However, in my opinion, the review article could have included additional details.

  1. Zr-89 is also used to trace other markers of T cells, for instance, CD2 and CD7 fab antibodies; these targets can be discussed in the review.

 We thank the Reviewer for their comments. We agree that a broader discussion of the use of Zr-89 for tracking of T cells, aside from our discussion in the context of animal studies, is certainly relevant given the topic of this review. In the aforementioned section we have added comments on Zr-89 Fab conjugates for CD2 and CD7, in addition to mentioning other targets that have been considered for use. 

  1. In addition, labeling specificity, bioavailability, and tissue distribution of listed labeling agents in Tables 1 &2 can be included. The advantages and limitations of these agents could be discussed.

We thank the reviewer for this suggestion. However, the main purpose of the review is to provide a broader overview of T-cell tagging, in order to present the evidence as it relates to gliomas, given this was an invited review about T-cell tagging in gliomas. Therefore, we had initially decided not to go into the specific pharmacodynamics of each of the labels, as not all of them are relevant to gliomas and brain imaging.

  1. T cells in the glioma section could be expanded. A table consisting of the T cell/T cell marker, labeling method, viability after labeling, functional properties of t cells, biodistribution, and anti-tumor properties for T cells in glioma sections can be included. Moreover, challenges in tracking T cells in glioma could be discussed.

The section on T-cells for glioma has been expanded to include additional details on each of the labels, with presentation of information, as available, as requested by the reviewer. We did not include a separate Table because in Tables 1 and 2, under “Cancer”, it pointed out which of the labels were specific to gliomas, and therefore, felt adding a separate table may add to redundant presentation of information. Additional challenges in tracking T cells in glioma have also been added to this section, which we thank the reviewer for, as it has improved the content of this section. 

  1. A reference list could be included for the existing Table 1 and Table 2.

 The references were added to Tables 1 and 2.

Reviewer 2 Report

The clinical application of T cell tracking for GBM is still in the evolution of new technology.  The directly monitor efficacy of immunotherapy is tested in patients with GBM.   Therefore, the advances in T cell tagging  can help noninvasively monitor activity in immunotherapy.  

The T cell compartment can form a powerful defense against extrinsic (e.g., pathogens) and intrinsic danger (e.g., malignant tumor cells). At the same time, specific subsets of T cells control this process to keep the immune system in check and prevent autoimmunity.  A wide variety in T cell functionalities exists, which is dependent on the differentiation and maturation state of the T cells.  Please discuss the application of flow cytometry in   comprehensive phenotyping of a T cell subset of interest.

These different subsets can be discriminated based on selective extracellular markers, in combination with intracellular transcription factor and/or cytokine stainings. Additionally, identification of very small subsets, including antigen-specific T cells, and important technical considerations of flow cytometry should be discussed. 

Author Response

The clinical application of T cell tracking for GBM is still in the evolution of new technology.  The directly monitor efficacy of immunotherapy is tested in patients with GBM.   Therefore, the advances in T cell tagging  can help noninvasively monitor activity in immunotherapy.  

The T cell compartment can form a powerful defense against extrinsic (e.g., pathogens) and intrinsic danger (e.g., malignant tumor cells). At the same time, specific subsets of T cells control this process to keep the immune system in check and prevent autoimmunity.  A wide variety in T cell functionalities exists, which is dependent on the differentiation and maturation state of the T cells.  Please discuss the application of flow cytometry in   comprehensive phenotyping of a T cell subset of interest.

These different subsets can be discriminated based on selective extracellular markers, in combination with intracellular transcription factor and/or cytokine stainings. Additionally, identification of very small subsets, including antigen-specific T cells, and important technical considerations of flow cytometry should be discussed. 

 We thank the review for their comments regarding the including of a discussion of the role that flow cytometry plays in phenotyping T cell subsets. We agree that this technique is integral to the characterization of T cells both generally, and in the context of identifying markers for T cell tracking in the context of diseases such as GBM. Thus, we have added to the section “T-cell labeling” a discussion of the role of flow cytometry in immunophenotyping T cell subsets, and in the importance of this technique in identifying meaningful markers for T cell tracking. We also touch on limitations of flow cytometry, and its capacity to identify rare subsets, including antigen-specific T cells. 

Reviewer 3 Report

The authors introduced the most update information about immunotherapy in glioblastoma. Overall, this article was clearly organized, well written, and very easy to understand. 

I only have two concerns. First, the total length seems pretty short, it's like a mini review rather than review to me. And the content focus on the T cell labeling and glioblastoma more separately than the real relationship between these two. I'll suggest the authors make changes to the title and leaving T cell labeling only in key words.

Author Response

The authors introduced the most update information about immunotherapy in glioblastoma. Overall, this article was clearly organized, well written, and very easy to understand.  

I only have two concerns. First, the total length seems pretty short, it's like a mini review rather than review to me. And the content focus on the T cell labeling and glioblastoma more separately than the real relationship between these two. I'll suggest the authors make changes to the title and leaving T cell labeling only in key words.

 We thank the Reviewer for their kind words regarding our submitted review. With regards to the length of this manuscript, we hope that additions made for the purpose of addressing reviewer comments have lengthened the review. With respect to the nature of the content, this was an invited review for novel imaging in brain tumors, but unfortunately, there is still sparse information on T cell labeling and glioma. We hope that our elaborated discussion under the section “T-cell Tracking in Gliomas,” as requested by Reviewer 3, has added additional information of T-cell tracking as it relates to gliomas.